# Outcomes of Thoracoscopic Lobectomy after Recent COVID-19 Infection

**DOI:** 10.3390/pathogens12020257

**Published:** 2023-02-05

**Authors:** Beatrice Leonardi, Caterina Sagnelli, Giovanni Natale, Francesco Leone, Antonio Noro, Giorgia Opromolla, Damiano Capaccio, Francesco Ferrigno, Giovanni Vicidomini, Gaetana Messina, Rosa Maria Di Crescenzo, Antonello Sica, Alfonso Fiorelli

**Affiliations:** 1Thoracic Surgery Unit, University of Campania Luigi Vanvitelli, 80131 Naples, Italy; 2Department of Mental Health and Public Medicine, University of Campania Luigi Vanvitelli, 80131 Naples, Italy; 3Pneumology Unit, Eboli Hospital; Eboli, 84025 Salerno, Italy; 4COVID-19 Hospital “M. Scarlato”, Department of Pneumology, 84018 Scafati, Italy; 5Anatomo-Pathology Unit, University Federico II, 80131 Naples, Italy; 6Department of Precision Medicine, University of Campania Luigi Vanvitelli, 80131 Naples, Italy

**Keywords:** COVID-19, oncology, surgery, thoracoscopic lobectomy, lung cancer

## Abstract

Background: The COVID-19 outbreak had a massive impact on lung cancer patients with the rise in the incidence and mortality of lung cancer. Methods: We evaluated whether a recent COVID-19 infection affected the outcome of patients undergoing thoracoscopic lobectomy for lung cancer using a retrospective observational mono-centric study conducted between January 2020 and August 2022. Postoperative complications and 90-day mortality were reported. We compared lung cancer patients with a recent history of COVID-19 infection prior to thoracoscopic lobectomy to those without recent COVID-19 infection. Univariable and multivariable analyses were performed. Results: One hundred and fifty-three consecutive lung cancer patients were enrolled. Of these 30 (19%), had a history of recent COVID-19 infection prior to surgery. COVID-19 was not associated with a higher complication rate or 90-day mortality. Patients with recent COVID-19 infection had more frequent pleural adhesions (*p* = 0.006). There were no differences between groups regarding postoperative complications, conversion, drain removal time, total drainage output, and length of hospital stay. Conclusions: COVID-19 infection did not affect the outcomes of thoracoscopic lobectomy for lung cancer. The treatment of these patients should not be delayed in case of recent COVID-19 infection and should not differ from that of the general population.

## 1. Introduction

The COVID-19 outbreak had a massive impact on healthcare worldwide. COVID-19 is an infectious disease caused by severe acute respiratory syndrome coronavirus 2 (SARS-CoV-2) first detected in December 2019 in Wuhan, (China) and declared a pandemic in March 2020 [1,2].

Oncological patients suffered many consequences of the COVID-19 pandemic. In the initial phases of the pandemic, the accessibility to diagnostic and therapeutic pathways was impaired due to the conversion of several departments into COVID-19 units, causing disruptions to oncological screening, treatment, and surveillance [3,4,5,6,7,8,9,10,11,12,13]. Additionally, COVID-19 infection can cause serious complications in oncological patients undergoing chemotherapy or recovering from recent surgery [14,15,16,17,18,19,20,21,22,23,24,25,26].

Lung cancer is nowadays one of the most common and fatal cancer and lobectomy is the treatment of choice for its curative intent [27,28,29,30,31,32,33,34,35,36,37,38,39,40,41]. Video-assisted thoracoscopic surgery (VATS) is currently the preferred technique for pulmonary lobectomy, being minimally invasive compared to open lobectomy.

It is fair to ask if COVID-19 is a risk factor for complications after VATS lobectomy, given that some patients, after the infection, present alterations in the pulmonary tissue, such as fibrosis and mediastinal lymphadenopathy, that may complicate the surgical dissection. In addition to that, long-term effects of COVID-19 infection, known as Long COVID [42,43,44,45,46,47,48,49,50,51,52], include the persistence of respiratory symptoms after COVID-19 infection.

The aim of this study was to evaluate the outcomes of thoracoscopic lobectomy for lung cancer in patients with recent COVID-19 infection, and whether previous COVID-19 infection was associated with a higher rate of complications and mortality compared to the control group.

## 2. Materials and Methods

### 2.1. Study Design

This was a retrospective mono-centric study including all consecutive patients undergoing VATS lobectomy for lung cancer between January 2020 and August 2022.

The clinical data of (i) patients undergoing VATS lobectomy for lung cancer and (ii) patients with complete data and a follow-up for at least 9 months were included in the analysis. We excluded the data of (i) patients undergoing lung resection different from lobectomy (i.e., wedge resection and segmentectomy); (ii) patients undergoing lobectomy via upfront thoracotomy; and (iii) patients with incomplete data and follow-up.

The endpoint of the study was to evaluate whether recent COVID-19 infection prior to surgery negatively affected the outcome of VATS lobectomy. The patients were divided into two groups based on whether they had a history of COVID-19 infection in the 3 months prior to surgery (the COVID group) or not (the no-COVID group).

Diagnosis of COVID-19 infection was documented in all patients of the COVID group with positive RT-PCR molecular swab test for SARS-CoV-2. Postoperative complications, mortality, and general surgical outcomes were evaluated and compared between the two groups. Furthermore, univariable, and multivariable analyses were performed to evaluate whether COVID-19 infection was an independent predictive risk factor for complications.

All procedures performed were in accordance with the international guidelines, with the Helsinki Declaration of 1975, revised in 1983, and the rules of the Italian laws of privacy. Each patient signed an anonymous informed consent letter for the use of their data for anonymous clinical investigations and scientific publications. The local research ethics committee approved the study design. Due to the retrospective nature of the study, no specific approval code was required because there was no modification in the standard of patient care.

### 2.2. Study Population

For each patient, the following data were recorded:Anamnestic data: demographic data (age, gender, body mass index (BMI)), comorbidities, smoking status, symptoms, laboratory data, respiratory function data, and tumor characteristics (histology and clinical stage), history of recent COVID-19 infection (in the 3 months preceding the surgery), and vaccination history against COVID-19.Peri and postoperative data: operative time (minutes), blood loss (mL), presence of pleural adhesions, conversion, need for transfusion, chest tube drainage output (mL), length of chest drainage (days), length of hospital stay (LHOS) (days), postoperative complications (including respiratory complications, cardiac complications, and other complications), and 90-day mortality.Pleural adhesions were classified according to the topography (in a portion of the pleural cavity or the majority of the pleural cavity) and according to the severity (loose or firm and vascular), as proposed by Kobayashi et al. [53].The complications were classified into 2 groups according to the Systematic Classification of Morbidity and Mortality After Thoracic Surgery [54]. Minor complications (grade I and II, requiring no therapy or pharmacologic intervention only) and major complications (grade IIIa, IIIb, IVa, and IVb, requiring surgical, endoscopic, or radiological intervention without general anesthesia, with general anesthesia, admission to the intensive care unit (ICU), and multiorgan failure, respectively).For patients in the COVID group, the severity of COVID-19 symptoms was recorded along with the volume of lung tissue impaired after COVID infection assessed on preoperative HRCT using the Simple Chest CT Severity Score [55]. The volume of lung tissue impaired was classified as score 0 (0%, none); score 1 (1–5%, minimal involvement); score 2 (6–25%, mild involvement); score 3 (26–49%, moderate involvement); score 4 (50–75%, severe involvement); and score 5 (≥75%, extensive involvement).

### 2.3. Surgical Procedure

In all cases, the tumor was staged with a whole-body PET/CT scan. Preoperative histological diagnosis was obtained in the majority of patients with CT-guided lung biopsy, or alternatively with a trans-bronchial needle aspiration biopsy or broncho-alveolar lavage during bronchoscopy. Neoadjuvant chemotherapy was administered when indicated following oncological consultation.

When the preoperative diagnosis was not possible or in case of undetermined results, the diagnosis was made through intra-operative pathological examination. Patients with lung function impairment (predicted postoperative FEV1 or DLCO < 30%) were evaluated with cardiopulmonary exercise testing (CPET) and if not functionally fit for surgery, redirected to alternative treatments through a multidisciplinary discussion.

For this reason, 10 patients were excluded from surgical treatment due to respiratory impairment: 2 patients for post-COVID-19 respiratory impairment and 8 for severe chronic obstructive pulmonary disease.

Twenty-four hours before admission, all patients routinely performed an RT-PCR molecular swab test for SARS-CoV-2, mandatory for hospital admission during the study period to exclude asymptomatic infections.

All the patients underwent pulmonary lobectomy with systematic ilo-mediastinal lymphadenectomy using an anterior thoracoscopic triportal approach. The procedure was performed under general anesthesia with selective intubation. At the end of the procedure, a single 28Fr drainage tube was placed in the pleural cavity through the camera incision and connected to an underwater seal chest drain system. The chest drainage was removed when re-expansion of the lung was achieved, when the amount of fluid drained was less than 250 mL in 24 h, and in absence of air leaks. After the surgery, the patients were directed either to follow-up or oncological treatment based on the pathological stage and histology.

The patients in the COVID group were scheduled for surgery after at least four weeks from reverse transcription polymerase chain reaction (RT-PCR) negativity, and in all cases performed an additional high-resolution CT scan of the thorax after COVID-19 infection. Patients with severe symptoms or that required hospitalization were carefully evaluated by the thoracic surgeon and anesthesiologist to schedule the surgery.

### 2.4. Statistical Analysis

Data were expressed as mean ± standard deviation (SD) for continuous variables and as absolute numbers and percentages for categorical variables. Intergroup differences were compared using chi-square for categorical variables and with a *t*-test for continuous variables. Univariable and multivariable logistic regression was performed to identify predictive risk factors for postoperative complications (dependent variables).

A *p*-value less than 0.05 was considered statistically significant. MedCalc statistical software (Version 12.3, Broekstraat 52; Mariakerke, Belgium) was used for this analysis.

## 3. Results

In the study period, 182 patients underwent lung resections for cancer. The data of 29 patients were excluded from the analysis due to resections different from lobectomy (*n* = 21); upfront thoracotomy (*n* = 6); and incomplete data (*n* = 2).

Thus, our study population counted 153 patients, 30 (19%) in the COVID group and 123 (80%) in the no-COVID group.

In the COVID group, sixteen (53%) patients had an asymptomatic COVID-19 infection, fourteen (47%) had symptomatic COVID-19, and two of them (7%) required hospitalization (Table 1).

The lung volume tissue involvement after COVID-19 was none (score 0) in five (17%) patients, minimal (score 1) in seven (23%) patients, mild (score 2) in eight (27%) patients, moderate (score 3) in seven (23%) patients, severe (score 4) in three (10%) patients, and extensive in none.

As summarized in Table 2, no significant differences were found between the two study groups regarding preoperative data, lobectomy type, and tumor characteristics, except for the BMI, which was higher in the COVID group compared with the no-COVID group (28.7 ± 4.1 vs. 26.3 ± 3.9, *p* = 0.0003).

The patients in the COVID group experienced more frequently thoracic pain as presenting symptoms compared to the no-COVID group (20% vs. 7%, *p* = 0.03).

There was no difference in the rate of vaccination against COVID-19 between groups (*p* = 0.36).

### Perioperative Outcomes and Complications

The data regarding perioperative outcomes and complications were summarized in Table 3.

No significant differences were found regarding operative time (*p* = 0.57), blood loss (*p* = 0.41), conversion (*p* = 0.82), and transfusion rate (*p* = 0.65) between the two groups. The chest drainage output (*p* = 0.16), length of chest drainage (*p* = 0.23), LHOS (*p* = 0.23), and ICU stay (*p* = 0.13) were also similar between groups.

The distribution of complications and postoperative outcomes were analyzed in the COVID group since during the study period different SARS-CoV-2 variants with different virulence were identified. Complications resulted uniformly distributed throughout the years, and postoperative outcomes were similar.

The COVID group patients had a higher incidence of pleural adhesions compared with the no-COVID group (46% vs. 21%, *p* = 0.006). The adhesions were more frequently firm and vascular and localized in the portion of the pleural cavity in both groups.

Overall, 59 patients (38%) had complications following the surgery. Of these, 49 patients had minor complications (32%) while 10 had major complications (6%).

Major complications consisted of respiratory failure (n = 2), myocardial infarction (n = 2), a pneumothorax that needed re-drainage (n = 2), pneumonia that requested ICU management (n = 2), empyema (n = 1), and bowel infarction (n = 1).

No significant difference was found regarding the incidence of postoperative complications between groups (*p* = 0.51). Mortality at 90 days was 2%, without significant differences between groups (*p* = 0.38).

The most common complications were respiratory (21%), mainly hypoxemia (20%), prolonged air leaks (18%), and atelectasis (18%).

The most common non-respiratory complication was postoperative anemia (16%). No patient needed reoperation, while two patients needed re-drainage due to persistent air leaks and atelectasis.

On univariable analysis (Table 4) capturing as a dependent variable the presence of complications, we found that significant risk factors were COPD (*p* = 0.015), coronary artery (*p* = 0.007), and FEV1 < 70% (*p* = 0.03).

However, in multivariable analysis, COPD (*p* = 0.006) and coronary artery disease (*p* = 0.016) were the only two independent significant prognostic factors for complications.

## 4. Discussion

The management of oncological patients during the COVID-19 pandemic has been challenging, and while it is suspected that COVID-19 infection during the perioperative period is a risk factor for morbidity and mortality [56,57,58], the majority of the studies in the literature are not specific to thoracic surgery or pulmonary lobectomy for lung cancer, or do not discriminate between preoperative and postoperative COVID-19 infection [59]. At present, only Gabryel et al. [60] evaluated the outcomes of anatomical lung resections in patients with COVID-19 history, reporting no difference in the incidence of postoperative complications and 90-day mortality between patients with and without COVID-19 history before surgery, except for a higher reoperation and re-drainage rate in the COVID group.

First, there was no difference in the incidence of complications and 90-day mortality between patients with a history of recent COVID-19 infection and patients without a history of recent COVID-19 infection, and we found that COVID-19 infection was not an independent predictive risk factor for complications. In our study population, we found that the patients with recent COVID-19 infection had more frequent pleural adhesions compared with the control group. Since pleural adhesions are often a result of previous inflammatory processes [61], this difference between groups is interesting and may be related to recent COVID-19 infection, but the studies that reported pleural adhesions at CT or thoracoscopy in patients with a history of COVID-19 are few, and are more often in patients with severe infection [62,63,64,65]. Despite the increased incidence of pleural adhesions in the COVID patients, there was no difference in the conversion rate between groups.

We reported two cases of extensive pleural adhesions in patients with a history of COVID-19 in Figure 1 and Figure 2; in both cases, the surgery was carried out using a thoracoscopy.

We also found that the patients with a history of COVID-19 had a higher rate of preoperative thoracic pain, which has been described as part of post-COVID pain syndromes [66] and should be investigated to exclude cardiac sequelae related to COVID.

Second, our results highlighted that coronary artery disease and COPD were risk factors for complications.

In the literature, COPD has been already associated with an increased risk for pulmonary complications [67,68], while coronary artery disease is not a recognized risk factor for complications after VATS lobectomy, reported only by some authors [69]. In our population, the patients with coronary artery disease mostly had cardiac complications, mainly arrhythmias that were managed in the immediate postoperative period with medical therapy.

Third, we observed that many patients were referred to our department exhibiting post-COVID chest X-ray or CT scans recommended by general practitioners as a follow-up after the infection. In these cases, COVID-19 infection has incidentally influenced the course of the patient’s treatment since small nodules could have otherwise gone unnoticed until a more advanced stage.

Post-COVID-19 HRCT often showed pulmonary changes, even in asymptomatic or mildly symptomatic patients, such as generally newly found GGO opacities, subpleural interstitial involvement, bronchial dilation, and subpleural bands. We performed a thorough examination of preoperative CT, and when needed we used 3D reconstruction of the lung to predict possible difficulties in the resection and to assist the nodule localization in case of lung tissue alterations.

The appropriate timing of surgery after COVID-19 is still debated; some authors suggest waiting 4–6 weeks to schedule elective surgery after COVID-19 infection [70], while others recommend waiting at least 7 weeks [71]. For patients with lung cancer, the timing of surgery is crucial and delays in surgical treatment can lead to pathological upstaging, which is associated with worse outcomes [72]. One of the predictors of upstaging is a delay in resection greater than 8 weeks [73], and between the duration of COVID-19 infection and the 4–7 weeks recommended to schedule elective surgery, these patients’surgery delays easily add up to 8 weeks.

The main limitations of our study is its retrospective nature and the relatively small sample size. Additionally, it is possible that some cases of COVID-19 were undetected, asymptomatic patients, but we accounted that this number would not be significant in our analysis since the patients were strictly monitored in the period before admission during the preoperative examination performed at our institution. Furthermore, the patients undergoing lung resection were highly selected and it could affect the surgical results. In the COVID group, we did not compare the complication rate between asymptomatic and symptomatic patients due to the small sample size and since the complication rate did not differ from that of the no-COVID group.

Our results point toward the safety of performing thoracoscopic lobectomy in patients with recent COVID-19 infection without lung function impairment and after at least four weeks from the infection. The treatment of patients with recent infection of COVID-19 should not be delayed, but it is mandatory to carefully evaluate the patient’s respiratory and cardiac conditions, in case of ICU admission for COVID-19 or severe respiratory/cardiac comorbidities. Performing a preoperative HRCT is useful to plan the surgery and predict difficulties in nodule localization and the dissection in case of lung tissue post-COVID alterations.

## 5. Conclusions

COVID-19 infection did not affect the outcomes of thoracoscopic lobectomy for lung cancer in patients without lung function impairment, including post-COVID consequences, and was scheduled after at least four weeks from the infection. The treatment of these patients should not be delayed in case of recent COVID-19 infection and should not differ from that of the general population. An appropriate cardiorespiratory and radiological post-COVID evaluation is mandatory to prevent complications.

## Figures and Tables

**Figure 1 pathogens-12-00257-f001:**
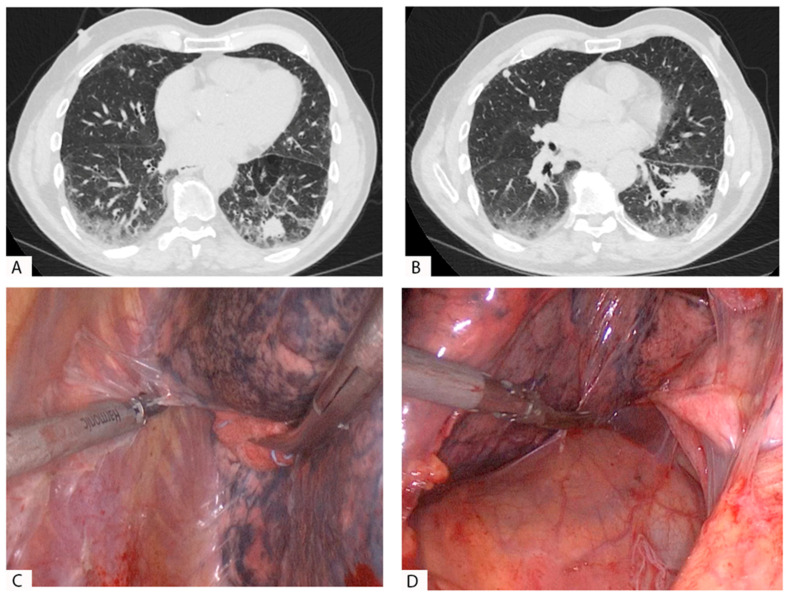
A patient with two localizations of adenocarcinoma in the left lower lobe (LLL) treated with VATS lobectomy. The patient had COVID-19 with mild symptoms 80 days before surgery. (**A**,**B**): Post-COVID HRCT of the thorax showing the nodules in the basal posterior and anteromedial segments of the LLL and multiple lung tissue alterations (ground-glass opacities, fibrosis, and bronchiectasis; simple chest CT severity score: 3). (**C**): Thoracoscopic view and dissection of pleural adhesions in the parietal pleura of the LLL. The adhesions were loose and in most of the pleural cavity. (**D**): Pleural adhesions in the mediastinal pleura.

**Figure 2 pathogens-12-00257-f002:**
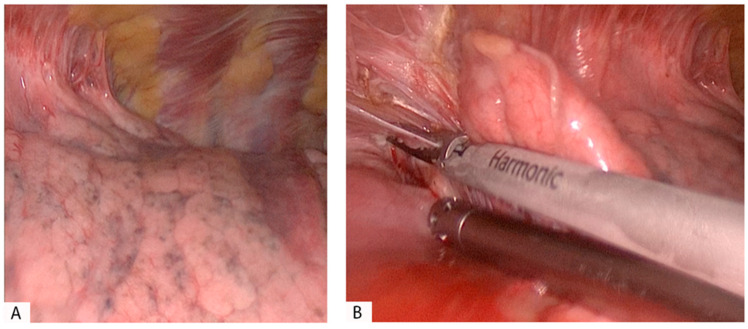
Patient with squamous adenocarcinoma in the right lower lobe (RLL) treated with VATS in the right lower lobectomy. (**A**,**B**): Thoracoscopic view and dissection of pleural adhesions in the parietal pleura of the RLL. The adhesions were firm and vascular, localized in a portion of the pleural cavity.

**Table 1 pathogens-12-00257-t001:** The severity of COVID-19 infection and the Simple Chest CT Severity Score for lung tissue involvement.

Severity of COVID-19 Symptoms	COVID Group (*n* = 30)
Asymptomatic, *n* (%)	16 (53)
Symptomatic (mild), *n* (%)	9 (30)
Symptomatic (moderate to severe), *n* (%)	3 (10)
Requiring hospitalization, *n* (%)	2 (7)
Requiring ICU admission, *n* (%)	0
**Simple Chest CT Severity Score**	
Score 0, (0%, no involvement)	5 (17)
Score 1 (1–5%, minimal involvement)	7 (23)
Score 2 (6–25%, mild involvement)	8 (27)
Score 3 (26–49%, moderate involvement)	7 (23)
Score 4 (50–75%, severe involvement)	3 (10)
Score 5 (≥75%, extensive involvement)	0

ICU: intensive care unit.

**Table 2 pathogens-12-00257-t002:** Characteristics of the study population.

Variables	Total (*n* = 153)	No-COVID (*n* = 123)	COVID (*n* = 30)	*p*-Value
Age (years), M ± SD	66.3 ± 7.7	66.8 ± 8.48	64.9 ± 7.95	0.28
Gender (male), *n* (%)	93 (61)	77 (62)	16 (53)	0.35
Smokers, *n* (%)	76 (50)	63 (51)	130 (43)	0.53
BMI (kg/m^2^), M ± SD	26.8 ± 4.01	26.3 ± 3.9	28.7 ± 4.1	0.0003
Neoadjuvant, *n* (%)	2 (1)	1 (1)	1 (3)	0.27
ASA physical status ≥ 3, *n* (%)	71(46)	58 (47)	13 (43)	0.7
Vaccination against COVID-19	99 (65)	81 (66)	18 (60)	0.36
Comorbidities, *n* (%): Diabetes COPD Hypertension Coronary artery disease History of cancer	24 (16) 43 (28) 104 (68) 25 (16) 50 (33)	21 (17) 37 (30) 84 (68) 21 (17) 40 (33)	3 (10) 6 (20) 20 (67) 4 (13) 10 (33)	0.34 0.27 0.86 0.62 0.93
Laboratory data, M ± SD: Hemoglobin (g/dL) White blood cells (/uL) Total protein (g/dL)	13.8 ± 1.6 7802.0 ± 2234.0 7.0 ± 0.7	13.8 ± 1.8 7725.0 ± 2100.0 6.9± 0.7	13.9 ± 1.2 8120.0 ± 2705.0 7.1 ± 0.6	0.62 0.38 0.27
Symptoms, *n* (%): Cough Thoracic pain Dyspnea Weight loss	57 (3) 15 (10) 31 (20) 4 (3)	48 (39) 9 (7) 24 (19) 4 (3)	9 (30) 6 (20) 7 (23) 0	0.36 0.03 0.64 0.31
Pulmonary function, M ± SD: FEV1 % DLCO % ppoFEV1 % ppoDLCO % 6MWT, meters	95 ± 19 89 ± 19 74 ± 16 70 ± 15 501 ± 116	94 ± 19 89 ± 20 74 ± 16 71 ± 16 502 ± 120	96 ± 20 86 ± 16 76 ± 16 68 ± 12 494 ± 84	0.58 0.47 0.48 0.39 0.82
Lobectomy type: RUL RML RLL LUL LLL	47 (31) 10 (7) 33 (22) 30 (20) 33 (22)	36 (29) 9 (7) 28 (23) 23 (19) 27 (22)	11 (36) 1 (3) 5 (16) 7 (23) 6 (20)	0.43 0.42 0.46 0.56 0.81
Pathological stage, *n* (%): IA IB IIA IIB IIIA	70 (45) 29 (19) 6 (4) 18 (12) 20 (13)	57 (46) 27 (22) 5 (4) 13 (11) 14 (11)	13 (43) 2 (7) 1 (3) 5 (17) 6 (20)	0.68 0.55 0.85 0.35 0.2
Histology, *n* (%): Adenocarcinoma Squamous cell carcinoma Typical carcinoid Atypical carcinoid LCNEC Others	92 (60) 32 (21) 5 (3) 6 (4) 3 (2) 15 (10)	73 (59) 25 (20) 4 (3) 5 (4) 3 (2) 13 (11)	19 (63) 7 (23) 1 (3) 1 (3) 0 2 (7)	0.68 0.71 0.58 0.85 0.25 0.52
Adjuvant chemotherapy	39 (25)	28 (23)	11 (36)	0.11

BMI: body mass index. COPD: chronic obstructive pulmonary disease. FEV1: forced expiratory volume in 1 s. DLCO: diffusing capacity of the lung for carbon monoxide. ppoFEV1: predicted postoperative FEV1. ppoDLCO: predicted postoperative DLCO. 6MWT: six-minute walking test; LCNEC: large cell neuroendocrine carcinoma. ASA physical status: American Society of Anesthesiologists physical status. RUL: right upper lobectomy. RML: right middle lobectomy. RLL: right lower lobectomy. LUL: left upper lobectomy. LLL: left lower lobectomy.

**Table 3 pathogens-12-00257-t003:** Perioperative outcomes and complications.

Variables	Total (*n* = 153)	No-COVID (*n* = 123)	COVID (*n* = 30)	*p*-Value
Operative time (minutes), M ± SD	285 ± 75	284 ± 76	292 ± 72	0.57
Pleural adhesions, n (%) Adhesion loose Adhesion firm and vascular In a portion of the pleural cavity In the majority of the pleural cavity	41 (27) 14 (9) 27 (18) 28 (18) 13 (8)	27 (21) 8 (6) 19 (15) 20 (16) 7 (5.7%)	14 (46) 6 (20) 8 (26) 8 (27) 6 (20)	0.006 0.01 0.01 0.01 0.01
Chest drainage (mL/day), M ± SD	140 ± 72	136 ± 71	157 ± 79	0.16
Chest tube duration (days), M ± SD	6.3 ± 4.3	6.5 ± 4.6	5.5 ± 2.3	0.23
Blood loss (mL), M ± SD	284 ± 60	285 ± 50	280 ± 61	0.41
Transfusion, n (%)	19 (12)	16 (13)	3 (10)	0.65
Conversion, n (%)	17 (11)	14 (11)	3 (10)	0.82
LHOS (days), M ± SD	7.5 ± 4.2	7.7 ± 4.5	6.7 ± 2.4	0.23
Complications, n (%): Patients with any complication Minor complications (grade I-II) Major complications (grade III-IV)	59 (38) 49 (32) 10 (6)	49 (40) 40 (32) 9 (7)	10 (33) 9 (30) 1 (3)	0.51 0.79 0.42
Respiratory complications, n (%): Patients with any respiratory complication Postoperative hypoxemia Prolonged air leaks Atelectasis Pneumonia Reintubation/prolonged intubation Re-drainage Respiratory failure Empyema Airway injury Bronchopleural fistula Hemothorax	32 (21) 31 (20) 28 (18) 18 (12) 9 (6) 5 (3) 2 (1) 2 (1) 1 (1) 0 0 0	27 (21) 27 (22) 25 (20) 16 (13) 8 (6) 5 (4) 1 (1) 2 (2) 1 (1) 0 0 0	5 (17) 4 (13) 3 (10) 2 (6) 1 (3) 0 1 (3) 0 0 0 0 0	0.52 0.29 0.19 0.33 0.26 0.5 0.54 0.48 0.62 / / /
Cardiac complications, n (%): Arrhythmias Myocardial infarction	5 (3) 2 (1)	5 (4) 2 (2)	0 0	0.26 0.48
Other complications, n (%): Subcutaneous emphysema Postoperative anemia Bowel infarction	19 (12) 25 (16) 1 (1)	15 (12) 20 (16) 1 (1)	4 (13) 5 (17) 0	0.37 0.65 0.62
ICU stay > 12 h, n (%)	17 (11)	15 (12)	2 (6)	0.13
90-day mortality, n (%)	3 (2)	3 (2)	0	0.38

ICU: intensive care unit; LHOS: length of hospital stay.

**Table 4 pathogens-12-00257-t004:** Univariable and multivariable analysis for complications (dependent variable).

Variables	Univariable		Multivariable	
	Odds Ratio	*p*-Value	Odds Ratio	*p*-Value
Age (<70 vs. >70)	0.52 (CI: 0.20–1.32)	0.17	-	-
Gender	1.65 (CI: 0.69–3.9)	0.25	-	-
Smoking status	1.80 (CI: 0.78–4.14)	0.16	-	-
Recent COVID-19	1.39 (CI: 0.59–4.10)	0.22	-	-
Diabetes	1.45 (CI: 0.52–18.41)	0.47	-	-
COPD	3.25 (CI 1.16–8.41)	0.015	5.82 (CI 1.42–14.22)	0.006
ASA ≥ 3	1.85 (CI: 0.46–7.35)	0.38	-	^-^
Coronary artery disease	3.60 (CI 1.41–9.13)	0.007	4.55 (CI: 1.32–13.82)	0.016
History of cancer	1.44 (CI 0.55–2.71)	0.10		
FEV1 < 70%	4.5 (CI: 1.20–16.41)	0.03		

COPD: chronic obstructive pulmonary disease; CI: confidence interval.

## Data Availability

Not applicable.

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
