# Peer review of "Outcomes of Thoracoscopic Lobectomy after Recent COVID-19 Infection"

_pathogens, 2023, doi:10.3390/pathogens12020257_

Round 1

Reviewer 1 Report

I thank the Editor for the opportunity of reviewing this paper.

The paper is interesting, but I believe that it may be improved as follows:   

1) Pleural adhesions were more frequent in the COVID group. Did pleural adhesions complicate the dissection in the COVID group?

2) Have you experienced difficulties in the tumor localization in the COVID group?

3) Why did you choose to focus on the outcomes of VATS lobectomy after recent COVID infection and not before surgery in general?

Author Response

Reviewer #1: The paper is interesting, but I believe that it may be improved as follows:   

Point 1:Pleural adhesions were more frequent in the COVID group. Did pleural adhesions complicate  the dissection in the COVID group?

Answer to the Reviewer point 1: The dissection with pleural adhesions was challenging in some cases, but generally there was no difference in operative times and conversion rate between the two study groups, so the adhesions have not influenced the surgical outcome.

Point 2: Have you experienced difficulties in the tumor localization in the COVID group?

Answer to the Reviewer point 2: The observation of the reviewer has been accepted and the new manuscript has been  modified accordingly. The preoperative CT was carefully examined, and in patients with fibrous alterations of the lung tissue we used 3D reconstruction to guide the dissection. We did not experienced difficulties in tumor localization in the COVID group.

Point 3: Why did you choose to focus on the outcomes of VATS lobectomy after recent COVID infection and not before surgery in general?

Answer to the Reviewer point 3: We chose to focus on patients with recent COVID infection since there is no consensus regarding the timing of surgery after COVID infection, and we wanted to investigate if patients with recent COVID had more complications than general population. There is one study that we reported in the discussion that analyzed outcomes of VATS lobectomy after COVID in general: “At present, only Gabryel et al. [60] evaluated the outcomes of anatomical lung resections in patients with COVID-19 history, reporting no difference in the incidence of postoperative complications and 90-day mortality between patients with and without COVID-19 history before surgery, except for a higher reoperation and re-drainage rate in the COVID group” (lines 230-234, page 7).

Reviewer 2 Report

In this manuscript the Authors analyze the results of a study designed to compare the results of VATS lobectomy for lung cancer in a group of 30 patients with a recent history of Covid-19 (in the previous three months) with a control group of 123 patients without previous Covid-19 infection, recruited from January 2000 to August 2022. The results of the study show that no differences in complication rate or 90-day mortality were observed between the two groups. However, patients with a recent Covid-19 infection had a higher incidence of pleural adhesions, although this finding was not associated with a higher conversion rate, drain removal time or hospital length of stay. The Authors therefore concluded that COVID-19 infection did not affect the outcomes of thoracoscopic lobectomy for lung cancer, and therefore surgical treatment of patients with recent infection of COVID-19 should not be delayed. The manuscript is based on a single-center retrospective study, but nevertheless the manuscript clearly addresses a specific topic as the role of VATS lobectomy in patients with previous Covid-19 infection and may be of interest.

Some minor issues should be addressed by the Authors:

-       Surgery in the group of patients with previous Covid-19 was scheduled after at least four weeks from the date of infection. This information should be stated in the conclusion section. 

-       The impact of Covid-19 varied noticeably during the phases of the pandemic according to the virulence of the different viral variants. It could be of interest to know if in the Covid-19 group the results varied along the different years of the study

-       The respiratory function of the patients included in the study was mostly within normal range  (see Table 2). The Authors should report if any patient was excluded from surgical treatment due to post-Covid-19 respiratory impairment. In fact, only two patients in the Covid-19 group required hospitalization during the infection

Author Response

Reviewer #2: In this manuscript the Authors analyze the results of a study designed to compare the results of VATS lobectomy for lung cancer in a group of 30 patients with a recent history of Covid-19 (in the previous three months) with a control group of 123 patients without previous Covid-19 infection, recruited from January 2000 to August 2022. The results of the study show that no differences in complication rate or 90-day mortality were observed between the two groups. However, patients with a recent Covid-19 infection had a higher incidence of pleural adhesions, although this finding was not associated with a higher conversion rate, drain removal time or hospital length of stay. The Authors therefore concluded that COVID-19 infection did not affect the outcomes of thoracoscopic lobectomy for lung cancer, and therefore surgical treatment of patients with recent infection of COVID-19 should not be delayed. The manuscript is based on a single-center retrospective study, but nevertheless the manuscript clearly addresses a specific topic as the role of VATS lobectomy in patients with previous Covid-19 infection and may be of interest.

Some minor issues should be addressed by the Authors:

Point 1: Surgery in the group of patients with previous Covid-19 was scheduled after at least four weeks from the date of infection. This information should be stated in the conclusion section. 

Answer to the Reviewer point 1: The observation of the reviewer has been accepted and the new manuscript has been  modified accordingly. We revised the conclusion section following your comment.

“Our results point towards the safety of performing thoracoscopic lobectomy in patients with recent COVID-19 infection without lung function impairment and after at least four weeks from the infection.” (changes in lines 300-302, page 9)

“COVID-19 infection did not affect the outcomes of thoracoscopic lobectomy for lung cancer in patients without lung function impairment, including post-COVID consequences, and scheduled after at least four weeks from the infection.” (changes in lines 309-311, page 9)

Point 2:  The impact of Covid-19 varied noticeably during the phases of the pandemic according to the virulence of the different viral variants. It could be of interest to know if in the Covid-19 group the results varied along the different years of the study.

Answer to the Reviewer point 2: The observation of the reviewer has been accepted and the new manuscript has been  modified accordingly. We analyzed the results of the COVID group during the different years of the study, and we found that there was no difference between the different years. We reported this data in the manuscript. Since the molecular swab test for SARs-CoV-2 did not identify the specific variants, it was not possible to investigate further the influence of SARs-CoV-2 variants.

“The distribution of complications and postoperative outcomes were analyzed in the COVID group since during the study period different SARS-CoV-2 variants with different virulence were identified. Complications resulted uniformly distributed throughout the years, and postoperative outcomes were similar.” (changes in lines 196-200, page 6)

Point 3:  The respiratory function of the patients included in the study was mostly within normal range  (see Table 2). The Authors should report if any patient was excluded from surgical treatment due to post-Covid-19 respiratory impairment. In fact, only two patients in the Covid-19 group required hospitalization during the infection

Answer to the Reviewer point 3: The observation of the reviewer has been accepted and the new manuscript has been  modified accordingly. We included data regarding the respiratory function evaluation process and patients excluded from surgical treatment due to respiratory impairment, including post-COVID impairment.

“Patients with lung function impairment (predicted post-operative FEV1 or DLCO < 30%) were evaluated with Cardiopulmonary Exercise Testing (CPET) and if not functionally fit for surgery, redirected to alternative treatments through multidisciplinary discussion.

For this reason, 10 patients were excluded from surgical treatment due to respiratory impairment: 2 patients for post-COVID impairment and 8 for severe chronic obstructive pulmonary disease.” (changes in lines 122-128, page 3)

Reviewer 3 Report

The authors present a study on the influence of recent COVID-19 infection in lung cancer patients on perioperative thoracoscopic lobectomy outcomes. In fact, the study design is interesting and in the current epidemiological situation this issue is at the top of clinicians interests, however, I also feel that some data should be clarified:

-        1. Only patients without lung function impairment are qualified for thoracic surgery. Even if this issue had been addressed by authors in discussion in line 258, I feel like the conclusions could mislead the readers. All patients with post-COVID lung impairment, even if the lung cancer stage allowed for operation, were successively disqualified from surgery. Therefore, in my opinion, the conclusions should be more cautious indicating that the history of recent COVID-19 in patients without lung function impairment (including post-SARs-CoV-2 consequences) does not influence on perioperative outcomes

2.   Please specify how COVID-19 had been identified? Only based on patients’ interview or with well documented positive SARs-CoV-2 swab result? This information is crucial knowing the fact that some patients recognize just a cold as a COVID-19 infection without the confirmation.

3. All patients prior to lung surgery had routinely performed swab test for COVID-19 to exclude asymptomatic infection? Please complete the manuscript with this information.

4. Did patients have a history of vaccination against COVID-19? This information should be added in the manuscript.

5. There is no information about volume of lung tissue impaired after COVID-19 (expressed in percentage), as all patients had HRCT prior to operation. This information could be important knowing the fact that there are discrepancies between clinical status of COVID infection and lung tissue involvement.  

6. As different lobes consist on different number of segments and therefore tissue volume, the information about the lobectomy type should be added. Were there any differences between COVID and non-COVID groups regarding the lobectomy type?

7. Patients at stage II and III of lung cancer were enrolled in the study. According to the guidelines, this group of patients should receive pre- and/or postoperative chemotherapy. Was it administrated? Could it have any influence on the results obtained? This information should be add in the manuscript.

Author Response

Reviewer #3: The authors present a study on the influence of recent COVID-19 infection in lung cancer patients on perioperative thoracoscopic lobectomy outcomes. In fact, the study design is interesting and in the current epidemiological situation this issue is at the top of clinicians interests, however, I also feel that some data should be clarified:

Point 1: Only patients without lung function impairment are qualified for thoracic surgery. Even if this issue had been addressed by authors in discussion in line 258, I feel like the conclusions could mislead the readers. All patients with post-COVID lung impairment, even if the lung cancer stage allowed for operation, were successively disqualified from surgery. Therefore, in my opinion, the conclusions should be more cautious indicating that the history of recent COVID-19 in patients without lung function impairment (including post-SARs-CoV-2 consequences) does not influence on perioperative outcomes

Answer to the Reviewer point 1: The observation of the reviewer has been accepted and the new manuscript has been  modified accordingly We modified the conclusions of the manuscript following your comment. We also specified in the methods the functional testing process for patients with lung function impairment in general.

“Our results point towards the safety of performing thoracoscopic lobectomy in patients with recent COVID-19 infection without lung function impairment and after at least four weeks from the infection.” (changes in lines 300-302, page 9)

“COVID-19 infection did not affect the outcomes of thoracoscopic lobectomy for lung cancer in patients without lung function impairment, including post-COVID consequences, and scheduled after at least four weeks from the infection.” (changes in lines 309-311, page 9)

“Patients with lung function impairment (predicted post-operative FEV1 or DLCO < 30%) were evaluated with Cardiopulmonary Exercise Testing (CPET) and if not functionally fit for surgery, redirected to alternative treatments through multidisciplinary discussion.

For this reason, 10 patients were excluded from surgical treatment due to respiratory impairment: 2 patients for post-COVID impairment and 8 for severe chronic obstructive pulmonary disease.” (changes in lines 122-128, page 3)

Point 2: lease specify how COVID-19 had been identified? Only based on patients’ interview or with well documented positive SARs-CoV-2 swab result? This information is crucial knowing the fact that some patients recognize just a cold as a COVID-19 infection without the confirmation.

Answer to the Reviewer point 2: The observation of the reviewer has been accepted and the new manuscript has been  modified accordingly COVID-19 infection has been identified through molecular swab test for SARs-CoV-2, since patient interview could be unreliable as you stated. We specified this information in the manuscript.

“Diagnosis of COVID-19 infection was documented in all patients of the COVID group with positive RT-PCR molecular swab test for SARs-CoV-2.” (changes in lines 75-77, page 2)

Point 3: All patients prior to lung surgery had routinely performed swab test for COVID-19 to exclude asymptomatic infection? Please complete the manuscript with this information.

Answer to the Reviewer point 3: The observation of the reviewer has been accepted and the new manuscript has been  modified accordingly. Yes, all patients prior to lung surgery routinely performed swab test for COVID-19 to exclude asymptomatic infection, as it was mandatory for hospital admission for surgery during the entire study period. We specified this information in the manuscript.

“Twenty-four hours before admission, all patients routinely performed RT-PCR molecular swab test for SARs-CoV-2, mandatory for hospital admission during the study period to exclude asymptomatic infections.” (changes in lines 129-131, page 3).

Point 4: Did patients have a history of vaccination against COVID-19? This information should be added in the manuscript.

Answer to the Reviewer point 4: The observation of the reviewer has been accepted and the new manuscript has been  modified accordingly. We added data regarding vaccination history against COVID-19 in Table 2 and in the results.

“There was no difference in the rate of vaccination against COVID-19 between groups (p=0.36). (changes in lines 175-176, page 4)

Point 5: There is no information about volume of lung tissue impaired after COVID-19 (expressed in percentage), as all patients had HRCT prior to operation. This information could be important knowing the fact that there are discrepancies between clinical status of COVID infection and lung tissue involvement.  

Answer to the Reviewer point 5: The observation of the reviewer has been accepted and the new manuscript has been  modified accordingly. We revised the manuscript including information about volume of tissue involvement at HRCT after COVID infection in the methods, results and in Table 1.

For patients in the COVID group, severity of COVID-19 symptoms was recorded along with volume of lung tissue impaired after COVID infection assessed on preoperative HRCT using the Simple Chest CT Severity Score [55]. The volume of lung tissue impaired was classified as score 0 (0%, none); score 1 (1-5%, minimal involvement); score 2 (6-25%, mild involvement); score 3 (26-49%, moderate involvement); score 4 (50-75%, severe involvement); score 5 (≥ 75%, extensive involvement).” (changes in lines 109-114, page 3)

“The lung volume tissue involvement after COVID-19 was none (score 0) in 5 (17%) patients, minimal (score 1) in 7 (23%) patients, mild (score 2) in 8 (27%) patients, moderate (score 3) in 7 (23%) patients, severe (score 4) in 3 (10%) patients, extensive in none.” (changes in lines 164-166, page 4)

Point 6: As different lobes consist on different number of segments and therefore tissue volume, the information about the lobectomy type should be added. Were there any differences between COVID and non-COVID groups regarding the lobectomy type?

Answer to the Reviewer point 6: The observation of the reviewer has been accepted and the new manuscript has been  modified accordingly. There were no differences regarding the lobectomy type between COVID and no-COVID group. We included this data in the results of the manuscript in Table 2.

“As summarized in Table 2, no significant differences were found between the two study groups regarding pre-operative data, lobectomy type and tumor characteristics (…)” (changes in lines 170-171, page 4)

Point 7: Patients at stage II and III of lung cancer were enrolled in the study. According to the guidelines, this group of patients should receive pre- and/or postoperative chemotherapy. Was it administrated? Could it have any influence on the results obtained? This information should be add in the manuscript.

Answer to the Reviewer point 7: The observation of the reviewer has been accepted and the new manuscript has been  modified accordingly. We reported the number of patients subjected to neoadjuvant and adjuvant chemotherapy (Table 2). We do not think that pre/post-operative chemotherapy influenced the outcome since it was equally distributed between groups and the complications were mainly recorded during the hospital stay and the early postoperative period, before chemotherapy treatment.

“Neoadjuvant chemotherapy was administered when indicated following oncological consultation.” (changes in lines 119-120, page 3)

“After the surgery, the patients were directed either to follow-up or oncological treatment based on the pathological stage and histology.” (changes in lines 138-140, page 3).